# Cost stickiness, absorbed slack and enterprise risks: Evidence from China

**Qian Binhua**[1]*, **Yang Boyuan**[2]

**1** Zhejiang Business Technology Institute, Ningbo, China, **2** University of International Business and Economics, Beijing, China

* herrgeld@126.com

## Abstract

This study delves into the complex domain of enterprise finance to explore the relationship between cost stickiness and enterprise risks, with a particular emphasis on the role of absorbed slack. Our analysis is based on a comprehensive examination of 3,177 Chinese listed enterprises over a period of ten years, from 2007 to 2022. The findings reveal that cost stickiness is a widespread phenomenon among Chinese listed enterprises, exhibiting variation across different industries. Notably, a strong positive correlation is identified between cost stickiness and enterprise risks, a correlation that remains consistent through various robustness tests, including indicator permutation, sample reconfiguration, and the resolution of endogeneity issues. The research further highlights the mediating effect of absorbed slack in the relationship between cost stickiness and enterprise risks. This study not only confirms the ubiquity of cost stickiness and its association with enterprise risks but also underscores the significant impact of absorbed slack. We offer a novel perspective on the interaction among cost stickiness, absorbed slack, and enterprise risks, providing valuable insights for enterprises aiming to enhance their risk management strategies. The conclusions and recommendations presented serve as a guide for those engaged in the challenging task of managing enterprise risks.

## 1. Introduction

In the complex realm of business administration, the interplay between cost stickiness, absorbed slack, and enterprise risks is crucial for financial robustness and operational effectiveness. Cost stickiness refers to the situation where costs do not decrease in proportion to a reduction in output levels. This phenomenon acts as a protective mechanism that preserves profit margins during economic contractions. It serves as a stabilizing force within the cost structure, enabling enterprises to withstand periods of diminished demand without facing the exacerbation of cost pressures.

Absorbed slack, on the other hand, is the strategic cushion that organizations employ to mitigate the impact of unforeseen circumstances. It is the buffer that allows for the absorption of shocks without compromising the core operations. By maintaining a level of flexibility in

**Data Availability Statement:** DOI: 10.6084/m9. figshare.25098803. (https://figshare.com/s/d96996b5b1300457cb17).

**Funding:** This paper is supported by the National Social Science Fund Project (22ATJ003), Zhejiang Province Social Science Planning Project

(24FNSQ078YB), and Ningbo Philosophy and Social Science Research Base Project (JD6-369), and no other external funding was obtained for this study.

**Competing interests:** The authors have declared that no competing interests exist.

resource allocation, businesses can adapt swiftly to changing market conditions, thereby preserving their competitive edge.

Enterprise risks represent the latent dangers that can undermine even the best-laid plans within an enterprise. These risks are critical differentiators between successful and unsuccessful outcomes, making their comprehension essential for maneuvering through the complex landscape of business. Effective risk management entails not only the identification and reduction of such risks but also the strategic use of these risks to foster innovation and drive business growth.

Together, cost stickiness, absorbed slack, and enterprise risks form a triad of factors that influence the trajectory of a business. They are the cornerstones upon which the edifice of enterprise sustainability is built. The relationship among these factors is dynamic and evolves with the business environment, making research into their interplay essential for understanding how businesses can optimize cost structures, manage resource allocation, and build robust risk management frameworks to ensure long-term viability and competitiveness. In an era where volatility is the new normal, mastering these concepts is not merely an option; it is a prerequisite for survival and success in the competitive landscape of global commerce.

In this paper, cost stickiness refers to asymmetric cost changes due to management decisions. Enterprise managers would like to retain idle resources when business volume declines to avoid adjustment costs. At the same time, when business volume rises, they tend to increase the necessary resources to sustain volume growth. In this way, it will result in the enterprise's costs increasing more when business volume grows than decreasing when business volume declines (Anderson et al., 2003) [1].

Depending on the degree of resource absorption the enterprise retains in the production process, we can categorize them into absorbed and unabsorbed slacks [2]. In this paper, absorbed slack refers to resources that have been absorbed into the organizational process, and those resources need to be used and reallocated.

Cost stickiness has impacts on business operations. Since the revenue and cost determine the enterprise's surplus, the cost stickiness will affect the enterprise's surplus characteristics to a certain extent, such as asymmetric timeliness and predictability. On the other hand, cost stickiness also affects enterprises' management decisions. Under the influence of enterprise cost management and decision-making, cost stickiness will have an impact on enterprise risks: cost stickiness reflects cost management behaviors, which may be a manifestation of the enterprise's possession of idle resources in order to positively respond to the market, and will reduce enterprise risks; whereas higher cost stickiness implies that the enterprise fails to cut idle resources promptly, which cures the asset allocation and reduces the quality of accounting information, and increases the enterprise risks. The impact of cost stickiness on enterprise risks is multifaceted. High cost stickiness can lead to increased earnings and asset volatility, raising the default and credit risks for the enterprise. It can also exacerbate the agency problem within an enterprise, where managers, influenced by self-interest, may lead business decision-making away from optimal resource allocation, increasing the cost stickiness level and consequently, the risk faced by the enterprise. Therefore, studying the impact and relationship between cost stickiness and enterprise risks has high theoretical and practical significance: it can help enterprises recognize the cost pattern and manage costs better, and also find ways to control risks by understanding the paths and drivers of cost stickiness affecting enterprise risks.

From the existing studies, scholars have carried out extensive researches on cost stickiness and enterprise risks and have mainly focused on the impact of cost stickiness on business management, including enterprise risks from the perspectives of adjustment costs, management expectations, and agency problems. [3].

The adjustment costs perspective suggests that when the adjustment cost of resources is significant, the enterprise managers prefer to retain unused resources to avoid this cost when business volume declines, and therefore, cost stickiness increases as the adjustment cost increases. It has been argued that there is a significant negative correlation between cost stickiness and enterprise value, where total costs, including sales costs, general costs, and management costs are sticky to varying degrees (Costa & Habib 2023 [4], Özkaya2021 [5]), those sold goods stickiness has a positive effect on the level of surplus management and administrative cost stickiness hurts the level of surplus management (Jia et al. 2022 [6]). There is a negative correlation between enterprise financialization and cost stickiness behavior (Zhou 2023 [7]). From the perspective of labor costs, the adoption of AI by enterprises increases the cost stickiness of the workforce. This effect is particularly significant for enterprises with a higher proportion of highly educated employees, a higher degree of capital intensity, and enterprises that reside in areas with a high degree of aging (Wang 2023 [8]). Enterprises with strong labor unions have higher labor cost stickiness and lower administrative cost stickiness (Chang 2022 [9]). Government audits have a positive effect on reducing labor stickiness in state-owned enterprises, and after being audited, enterprises with excess employees and labor cost stickiness are significantly lower (Li 2023 [10]). For enterprises with high labor cost stickiness, media reports can effectively improve labor investment efficiency (Liu 2023 [11]).

Regarding external institutional effects, a solid political system is associated with high cost stickiness (Kuo 2023 [12]). Effective tax avoidance mitigates tax stickiness, but R&D investment offsets this effect (Sun 2023 [13]). In emerging markets, market competition can reduce cost stickiness (Li 2021 [14]). In addition, enhancing infrastructure development represented by high-speed rail, accepting investments from mutual funds, conducting credit default swaps (CDS), and external analysts' forecasts for conducting enterprises with accurate and diversified forecasts can reduce enterprises' labor cost stickiness (Zhou 2022 [15], Wong 2022 [16], Dai 2023 [17], Mohammed et al. 2021 [18]).

The management expectations perspective suggests that when managers feel optimistic or pessimistic about future demand, it will affect an enterprise's cost stickiness. When optimistic about future demand, they are more likely to retain idle resources when sales decline because they expect to use them after demand rebounds. This tendency to retain resources will increase cost stickiness. When managers feel pessimistic about future demand, they may aggressively cut idle resources when sales decline because they expect these resources to remain idle. This tendency to cut resources will lead to decrease cost stickiness. From the perspective of managers' characteristics, if CEOs and CFOs are overconfident, it will lead to high cost stickiness (Chen 2022 [19], Hur 2019 [20], Lai 2021 [21]). Whether the CEO has military experience or not also affects cost stickiness. Enterprises whose CEOs have been in the military have high cost stickiness than enterprises whose CEOs have not been in the military (He 2023 [22]). The CEO's hometown plot is significantly associated with the enterprise's cost stickiness, and enterprises headquartered in the CEO's hometown tend to have higher cost stickiness (Long 2023 [23]). Higher gender diversity and better co-CEO systems in enterprises are associated with lower cost stickiness (Le 2022 [24], Lee 2019 [25]). From the perspective of management expectations, cost stickiness tends to increase when managers are optimistic about future economic performance or when managers have high-risk appetites (Han 2020 [26], Li 2020 [27]). The higher the management execution efficiency, the higher the enterprise's cost stickiness (Zhai 2023 [28]). Management expectations will significantly impact cost stickiness when adjustment costs and unused resources are high (Chen 2019 [29]). CEOs tend to lead to inefficient labor investments when they are close to independent board members (Khedmati 2020 [30]). In terms of the external environment on management expectations effects, local official turnover reduces the cost stickiness of local enterprises (Jian 2023 [31]), political uncertainty

caused by elections increases enterprises' cost stickiness (Lee 2020 [32]), while religious beliefs ease enterprises' cost stickiness (Ma 2021 [33]). External political relations, as well as law and regulation making, will increase enterprise cost stickiness (Garkaz 2019 [34], Kim 2020 [35], Ranjan 2022 [36]).

The agency problem perspective suggests building business empires. Managers will invest excess resources when sales are growing and are unwilling to cut idle resources when sales decline. Such resource decisions lead to cost stickiness [37, 38]. From the perspective of management motivation, managers' customer orientation and employee orientation lead to higher cost stickiness(Liu 2019 [39], Hartlieb 2023 [40]). Opportunistic expenditures due to managerial free-riding behavior before an enterprise's merger or acquisition increase cost stickiness (Nagasawa 2021 [41]). Cost stickiness is significantly negatively related to the acquirer's abnormal returns after the M&A announcement (Uğurlu 2019 [42]). Implementing dividend payments and stock buybacks decreases cost stickiness (Smith 2021 [43]). Cost stickiness is prominent when public school principals face high enrollment pressures (Khan 2020 [44]). From a free cash flow perspective, internal financing constraints and liquidity from excess IPO financing significantly contribute to cost stickiness. In contrast, debt financing constraints and equity financing constraints inhibit cost stickiness significantly (Chen 2021 [45], Koo 2021 [46]). Enterprises participating in PPP projects have higher cost stickiness (Qing 2023 [47]), while the appreciation of enterprises' real estate helps reduce cost stickiness (Li 2023 [48]). From the perspective of the strength of social norms, some scholars argue that the greater the social responsibility of the enterprise, the more pronounced the labor cost stickiness (Eun 2023 [49]), but some scholars believe that enterprise social responsibility plays an inhibiting role on cost stickiness (Ma 2023 [50] Xu 2023 [51]), such as investors' on-site visits to listed enterprises can inhibit enterprises' cost stickiness (Yao 2023 [52]). The higher the cost stickiness of an enterprise, the lower the earnings transparency of the enterprise (Oh 2021 [53]).

As a summary of the existing studies, scholars have mainly studied cost stickiness as an essential manifestation of enterprises' cost management behavior. At the same time, there are fewer studies on the economic consequences and influence mechanisms of cost stickiness. For example, different influencing factors may lead to an increase or decrease in cost stickiness, but their increase or decrease does not necessarily lead to worse or better economic consequences.

Compared with existing studies, our study contributes in the following aspects. First, based on the data of Chinese A-share non-financial listed enterprises, we analyze the prevalence of cost stickiness in enterprises. Second, we investigate the impacts and the positive correlations between cost stickiness and enterprise risks. Finally, along the path of absorbed redundancy, we study the mechanism of cost stickiness on enterprise risks, and absorbed slack has a mediating effect that affects cost stickiness and enterprise risks.

The remainder of this paper is assigned as follows. Section 2 provides theoretical analysis and research hypotheses. Section 3 describes the research design and data source. Section 4 analyzes the empirical results, including descriptive statistics, industry distribution, correlation test, multiple regression analysis, endogeneity test, and robustness tests. We also conduct an intermediate effect test. Section 5 gives discussion and Section 6 gives conclusion, implication and limitation.

## 2. Theoretical analysis and research hypothesis

This section theoretically analyzes and compares the relationships among cost stickiness, absorbed slack, and enterprise risks.

## 2.1 Cost Stickiness and enterprise risks

Managers tend to cut idle resources less even when the level of business volume decreases, resulting in a lower reduction in the amount of costs than in the volume of business, thus manifesting itself as cost stickiness. Cost stickiness increases enterprise risks, including operational risk and financial risk. Among them, operational risk is the operating profit change caused by the uncertainty of the production and operation of the enterprise; financial risk refers to the change in the enterprise's financial position due to the impact of various uncertainties in its financial activities, such as fund-raising.

First, cost stickiness increases the risk of business operations. Cost stickiness means that the enterprise fails to cut idle resources in time when the business volume declines, and these idle, absorbed resources are challenging to reallocate in the subsequent production process, which makes it difficult for the enterprise to adjust the cost in time according to the changes of internal and external environments [54]. These absorbed idle resources are often represented by low liquidity assets such as idle and unprocessable specialized equipment that occupy costs. On the one hand, they will increase the holding cost; on the other hand, they will solidify the enterprise development path and increase the enterprise cost of transformation and upgrading. In addition, asymmetric cost changes also prevent enterprise management from seizing suitable investment opportunities, affecting management's short-term decisions and thus increasing enterprise risks [55]

Second, cost stickiness increases enterprise financial risk. Asymmetric cost changes and enterprise volume lead to the asymmetry between surplus and revenue, decreasing the accuracy of enterprise surplus forecasts [56], and the quality of accounting information decreases, thus pushing up the cost of debt financing and the cost of equity enterprises' finance. On the one hand, under cost stickiness, the absorbed idle resources are internalized in the production process, which will lead to the fluctuation of the enterprise's non-payment cost, superimposed on the fluctuation of the enterprise's operating profit, and also lead to the increase of the volatility of the enterprise's cash flow, then increase the volatility of the assets [57]. When the enterprise's asset value is lower than its default value, the higher the volatility and the default probability, the higher rate the credit risk appear [58]. On the other hand, the absorbed idle resources under cost stickiness reduce enterprises' profits while decreasing enterprises' surplus predictability and robustness [59, 60], increasing the information risk of equity investors. Combining these two factors will increase the cost of debt financing, equity financing, the likelihood of changes in the enterprise's financial position, and the enterprise's financial risk.

Third, the positive relationship between cost stickiness and enterprise risks can be further understood through the following mechanisms. Firstly, the earnings volatility. Enterprises with high cost stickiness will experience greater volatility in earnings due to their inability to adjust costs in response to changes in demand. This earnings volatility can lead to a higher risk of financial distress and bankruptcy. Secondly, the investment decisions. Cost stickiness can influence investment decisions, as enterprises may be more cautious about investing in new projects or expanding operations due to the high fixed costs associated with such endeavors. This conservatism can limit growth opportunities and increase the enterprises' risk profile. Thirdly, the credit ratings and financing costs. Enterprises with high cost stickiness may have a lower credit rating due to the perceived higher risk of financial distress. This can result in higher financing costs, which further increases the enterprise's overall risk.

Fourth, enterprises exhibit a tendency to cluster, which in turn influences the degree of cost stickiness. According to Subramanian and Weidenmier (2003) [61], there are significant industry differences in cost stickiness, with factors such as low/high competitiveness. This suggests that when developing hypotheses regarding cost stickiness, the clustering nature of

enterprises and the varying degrees of competitiveness within different industries should be integral components.

The above theoretical analysis suggests that cost stickiness is indeed positively related to enterprise risks. The inability to adjust costs in response to changes in demand can lead to greater earnings volatility, more conservative investment decisions, higher financing costs. These factors collectively contribute to an increased risk profile for enterprises with high degrees of cost stickiness. Managers and stakeholders should be aware of this relationship and consider strategies to mitigate cost stickiness, such as adopting more flexible cost structures, to reduce enterprise risks and enhance the enterprises' long-term viability.

The theoretical analysis leads to hypothesis $H_1$.

$H_1$: Cost stickiness is positively related to enterprise risk, and enterprises in different industries have different cost stickiness.

## 2.2 Mediating role of absorbed slack

Under the role of cost stickiness, when the enterprise volume declines, the managers still tend to maintain the original enterprise scale and the lower capacity of resources, resulting in the enterprise costs and the enterprise volume showing asymmetric changes, which will solidify the enterprise resource allocation, reduce the quality of accounting information, push up the operational and financial risks, and subsequently increase the enterprise risks.

This paper argues that absorbed slack mediates the relationship between cost stickiness and enterprise risks for the following reasons.

First, absorbed slack is less liquid and flexible than unabsorbed slack [62]. For reasons of adjustment costs and other considerations, such as employees failing to dismiss due to termination costs and equipment that they fail to dispose of promptly due to disposal costs, enterprises will retain specific idle resources when business volume declines, leading to cost stickiness. These idle resources, which are already less liquid and less flexible, are gradually absorbed by the enterprise during production and operation. Absorbed slack usually has a high degree of resource specialization and low flexibility, and also has limited applicability to specific situations [63]. When an enterprise faces environmental changes, it is difficult for the absorbed slack to quickly adapt to the new demand, which will take up a large amount of the enterprise's cost and solidify its resource allocation, thus increasing the enterprise's risks.

Second, absorbed slack is less identifiable than unabsorbed slack and is more difficult for managers to recognize and exploit [64]. When an enterprise fails to cut down some resources, these resources are gradually absorbed by the enterprise and become absorbed slack with lower recognizability. For example, processed material inventory goes from "raw materials" to "production costs" in accounting accounts, and lower production capacity is reflected by lower gross profit margin, which cannot be directly recognized in the general ledger. In contrast, lower production capacity is reflected in lower gross margins, which are hidden and cannot be recognized directly in the general ledger. When managers cannot recognize the absorbed slack, they will tend to reduce the financial statements, and also diminish the relevance and quality of the accounting information, pushing up the cost of debt and equity financing and increasing the enterprise's financial risk. On the contrary, unabsorbed slack is more recognizable. The adjustments of unabsorbed slack can be reflected, and decision-making can be adjusted promptly in the financial statements, which will not increase the financing cost of the enterprise [65]. On the other hand, unabsorbed slack does not increase the financing cost and financial risk of the enterprise because it is recognizable, and the adjustments can be reflected in the financial statements on time so that managers can make timely decisions and adjustments.

Third, this mediation can be analyzed through the following theoretical pathways. Firstly, the resource inefficiency. When costs are sticky, enterprises may maintain higher levels of resources than necessary to meet current demand. This excess capacity represents absorbed slack, as these resources are not contributing to revenue generation but still incur costs. The inefficiency of resource utilization can lead to higher operational costs and reduced profitability, thereby increasing the enterprise's financial risk. Secondly, the reduced flexibility. High levels of absorbed slack can reduce an enterprise's operational flexibility. With more resources tied up in unproductive assets, the enterprise may be slower to respond to market changes or to adjust its operations to align with shifting demand patterns. This lack of agility can make the enterprise more vulnerable to market shocks and competitive pressures, thus elevating enterprise risks. Thirdly, the opportunity costs. The presence of absorbed slack implies that an enterprise is not fully leveraging its resources, leading to opportunity costs. These unexploited resources could have been used to generate additional revenue or to invest in growth opportunities. The foregone benefits from these missed opportunities can indirectly increase the enterprise's risk by limiting its growth potential and competitive advantage.

The theoretical analysis supports the hypothesis that absorbed slack mediates the relationship between cost stickiness and enterprise risks. By understanding and addressing the factors that contribute to absorbed slack, enterprises can mitigate the risks associated with cost stickiness and enhance their overall risk management capabilities. Based on this, this paper proposes hypothesis H2.

H2: Cost stickiness increases enterprise risks by increasing the enterprise's absorbed slack, i.e., absorbed slack has a mediating effect that affects the relationship between cost stickiness and enterprise risks.

Fig 1 gives the theoretical framework cost stickiness, absorbed slack and enterprise risks.

## 3. Research design

### 3.1 Sample and data collection

In this paper, we use Chinese A-share listed enterprises from 2007–2022 as the research sample and clean the raw data according to the following criteria: (1) exclude the sample of the financial enterprises; (2) exclude the sample of ST enterprises; (3) exclude the sample of missing relevant data; (4) exclude the sample of return on assets that is not continuous in three years (year t-2 to year t); and (5) perform two-side 1% Winsorize shrinkage treatment for all continuous variables. After collation, we obtained 3,177 enterprises from 2007–2022.

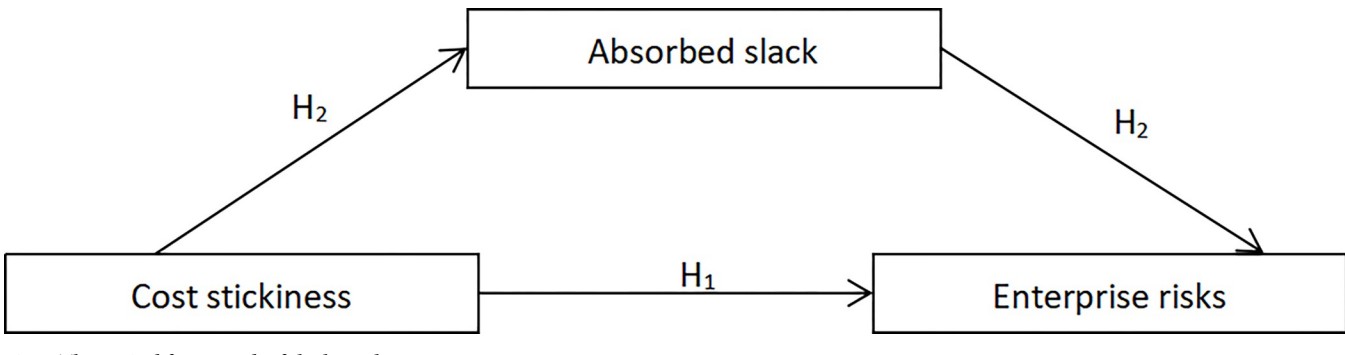

**Fig 1. Theoretical framework of the hypotheses.**

We obtained all the data from the CSMAR(China Stock Market & Accounting Research) database. This database is a comprehensive financial database widely recognized for its extensive coverage of Chinese financial markets. It encompasses a broad spectrum of data including stock market information, accounting data, economic indicators, and corporate governance details. The database is known for its rigorous data collection and verification processes, ensuring high data quality and reliability. Its effectiveness is underscored by its widespread use in academic research and professional analyses, making it a trusted source for financial data in China. The authority of CSMAR is further bolstered by its alignment with international standards and its commitment to transparency and accuracy, solidifying its position as a leading database in the field. We used Stata 16.0 software to process the data.

## 3.2 Variables

**3.2.1 Explained variable.**　The explained variable is enterprise risks. Referring to Ball et al. [66], We chose the volatility of accounting earnings ($ROAVol_{i,t}$) to measure enterprise risks. The volatility of enterprises' accounting earnings reflects enterprises' operational risk as well as their financial risk. Industry adjustments are used to mitigate the impact of differences between industries. It is calculated as the standard deviation of the industry-adjusted enterprise return on assets from year t-2 to year t. The industry-adjusted return on assets is the return on assets of the enterprise for the year minus the average of the return on assets of the industry in which the enterprise operates for the year.

**3.2.2 Explanatory variables.**　Referring to Weiss [56], Eq (1) is used to measure the cost stickiness of enterprise i in year t:

$$Sticky_{i,t} = \ln(\frac{\Delta Cost}{\Delta Sale})_{i,t,\omega1} - \ln(\frac{\Delta Cost}{\Delta Sale})_{i,t,\omega2} \tag{1}$$

where $\omega_1$ and $\omega_2$ denote the quarter closest to the end of the year in which the revenue of enterprise i rises and falls, respectively, in year t. Sale and Cost denote the value change in operating revenues and total costs in the quarter of $\omega_j$ (j = 1,2) in year t for enterprise i. The total cost is calculated as the sum of operating, selling, and administrative expenses in the enterprise's income statement. By covering the cost accounts mainly involved in the production process, the asymmetric relationship between the total cost and enterprise volume can be more accurately portrayed, and the measurement error caused by the misclassification of cost accounts can be avoided. The cost stickiness $Sticky_{i,t}$, obtained by this formula, indicates the cost stickiness of enterprise i in year t. If the value is positive, it means that the cost increase due to revenue increase in that year is more significant than the decrease in cost due to a decrease in revenue, and there is cost stickiness in the enterprise. The larger the value, the greater the cost stickiness. The formula takes the opposite number under the Weiss calculation so that the variable changes in the same direction as the degree of cost stickiness of the enterprise and can interpret the regression results more intuitively.

To illustrate that the cost stickiness indicator constructed in Eq (1) is a better proxy variable for the degree of enterprises' cost stickiness, this paper uses enterprises' asset density ($AINT_{i,t-1}$), GDP growth rate ($\Delta GDP_{t-1}$), and free cash flow ($FCF_{i,t-1}$) [37, 38] as proxy variables for adjustment costs, management expectations, and agency problems to test the validity of cost stickiness. The reason for the lagged treatment of these three indicators is that these three proxy variables are calculated using year-end values. In contrast, the cost stickiness indicator represents the degree of cost stickiness of the enterprise throughout the year, and the treatment afterward avoids the endogeneity of mutual causation.

**3.2.3 Control variables.**　Referring to Ball et al. [66] equity concentration ($Shrz_{i,t}$), executive compensation ($Com_{i,t}$), the proportion of independent directors ($Indratio_{i,t}$), enterprise

size ($Size_{i,t}$), gearing ratio ($Lev_{i,t}$), growth ($Growth_{i,t}$), and listing age($Age_{i,t}$) are chosen as the control variables in the regression of cost stickiness and enterprise risks.

Equity Concentration: Equity concentration is a quantitative indicator of whether all shareholders are concentrated or dispersed regarding the shareholding proportion. Shareholding concentration is an essential indicator of the state and stability of an enterprise's shareholding distribution. It is expressed as the ratio of the shareholding of the second largest shareholder to that of the first largest shareholder.

Executive Compensation: Executive compensation refers to monetary and non-monetary compensation received by enterprise executives over a certain period, including base salary, performance bonuses, allowances, stock incentives, and welfare benefits. It is expressed as the total remuneration of the top three directors and supervisors and treated logarithmic.

The proportion of independent directors: According to the opinions on the reform of the independent director system of listed enterprises in China issued in 2023, the number of independent directors as an essential person for checking and balancing the non-independent directors and balancing the diversity of the enterprise's management, shall not be less than one-third of the members of the board of directors, and shall include at least one professional accountant. The number of independent directors should account for more than half of the Board of Directors' committee members, such as the Audit Committee, the Nomination Committee, the Compensation and Evaluation Committee, and the Strategy Committee. They were expressed as the ratio of the total number of independent directors to the total number of board members.

Enterprise size: According to the relevant standards and regulations, enterprise size can be categorized as extra-large, large, medium, small, and micro. It is expressed in terms of total enterprise assets and treated in logarithms.

Gearing ratio: The gearing ratio is used to measure the ability of an enterprise to utilize funds provided by creditors for its operations and to reflect the degree of security of creditors in granting loans. It is expressed as total liabilities divided by total assets.

Growth: Growth refers to the ability of an enterprise to increase its value-added continuously and to add value to the enterprise, both in terms of long-term profitability and in terms of the continuous appreciation of the enterprise's assets, including intangible assets. It is expressed in terms of the growth rate of operating income.

Listing age: Listing age is the number of years an enterprise has been listed on the stock exchange, expressed as the number of years between the year of incorporation and the year of listing.

**3.2.4 Intermediate variable.** Referring to Bradley et al. [67], the ratio of marketing and administrative expenses to operating revenues is used, and the industry average value is deducted for industry adjustment. The larger the indicator, the more slack has been absorbed, i.e., the more idle resources in the enterprise that are difficult to reallocate and identify.

Table 1 gives the definitions of the main variables used in the model.

## 3.3 Models

This paper constructs Eq (2) to test the validity of cost stickiness:

$$Sticky_{i,t} = \beta_0 + \beta_1 AINT_{i,t-1} + \beta_2 \Delta GDP_{i,t-1} + \beta_3 FCF_{i,t-1} + Year + Industry + \varepsilon_{i,t} \qquad (2)$$

To avoid endogeneity, this paper tests the validity of cost stickiness using the lagged terms of the explanatory variables. $\beta_1$, $\beta_2$, and $\beta_3$ reflect the effects of adjustment costs, management optimism, and agency problems on cost stickiness. According to the theoretical analysis in the previous section, all these three coefficients should be significantly positive. $\varepsilon_{i,t}$ is the random disturbance term.

**Table 1. Definition of main variables.**

| Variable name | Variable symbol | Variable Definition |
|---|---|---|
| Enterprise risks | $ROAVol_{i,t}$ | The standard deviation of industry-adjusted return on assets from year t-2 to year t |
| Cost stickiness | $Sticky_{i,t}$ | Referring to Weiss, see Eq (1). |
| Asset density | $AINT_{i,t-1}$ | Total assets/operating income |
| Free cash flow | $FCF_{i,t-1}$ | (Net cash flows from operating activities—cash paid in dividends, profits or interest payments)/total assets |
| Equity concentration | $Shrz_{i,t}$ | The ratio of shareholding of the second largest shareholder to the first largest shareholder of the enterprise |
| Executive remuneration | $Com_{i,t}$ | Logarithmic total remuneration of the top three directors and supervisors |
| The proportion of independent directors | $Indratio_{i,t}$ | Number of independent directors/number of total directors |
| Enterprise size | $Size_{i,t}$ | Total assets in logarithms |
| Gearing ratio | $Lev_{i,t}$ | Total liabilities/total assets |
| Growth | $Growth_{i,t}$ | Revenue growth rate |
| Listing age | $Age_{i,t}$ | Year—Year of Listing + 1, in logarithms |
| Absorbed slack | $Abslack_{i,t}$ | Industry-adjusted marketing and administrative expenses to operating revenues |
| Equity incentive | $EI_{i,t}$ | If the management shareholding is more significant than the annual industry median, it takes 1. Otherwise, it takes 0. |
| Year effect | Year | If the sample belongs to that year, it takes 1. Otherwise, it takes 0. |
| Industry effect | Industry | If the sample belongs to the industry, it takes 1. Otherwise, it takes 0. |
| | | All industries use one code except for manufacturing, which uses two. |

Referring to Wen et al. [68] this paper tests the mediating effects as follows. Fig 2 shows the specific testing process, with the coefficients a, b, c, and c' given by Eqs (3) through (5).

$$ROAVol_{i,t} = cSticky_{i,t} + \beta_k Controls_{i,t} + Year + Industry + \varepsilon_{i,t} \qquad (3)$$

$$Abslack_{i,t} = aSticky_{i,t} + \beta_k Controls_{i,t} + Year + Industry + \varepsilon_{i,t} \qquad (4)$$

$$ROAVol_{i,t} = c'Sticky_{i,t} + bAbslack_{i,t} + \beta_k Controls_{i,t} + Year + Industry + \varepsilon_{i,t} \qquad (5)$$

If cost stickiness increases enterprise risks, the coefficient on $\beta_1$ should be significantly positive. $\beta_k$ reflects the effect of the control variable on enterprise risks. If $\beta_k$ is greater than 0, the control variable increases enterprise risks; if $\beta_k$ is less than 0, the control variable decreases enterprise risks.

We must consider the significance and positive and negative coefficients a, b, and c in testing the mediation effect. Suppose all three coefficients are significantly positive, absorbed slack mediates between cost stickiness and enterprise risks, cost stickiness increases enterprise risks by increasing the absorbed slack of the enterprise.

## 4. Empirical results and analysis

In this section, we conduct empirical analyses, including descriptive statistics, industry distribution, correlation tests, multiple regression analyses, endogeneity tests, and robustness tests.

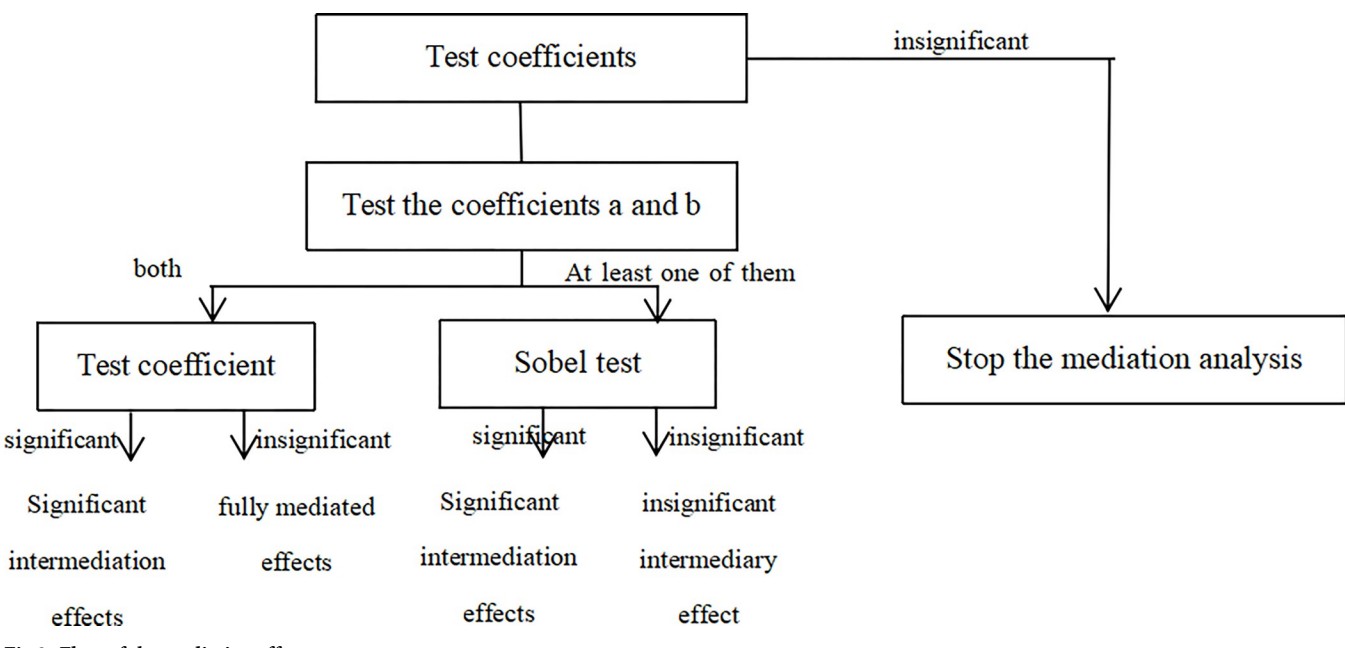

**Fig 2. Flow of the mediating effect test.**

### 4.1 Descriptive statistics

Table 2 reports the descriptive statistics of the main variables. The explained variable enterprise risks($ROAVol_{i,t}$) has a mean of 0.031 and a standard deviation of 0.037. The explanatory variable cost stickiness ($Sticky_{i,t}$) has a mean value of 0.176 and a median value of 0.097, suggesting that enterprises are generally characterized by varying cost stickiness. In addition, some enterprises are characterized by inverse cost stickiness, i.e., enterprises' costs increase less with the growth of enterprises' revenues than decrease with the decrease of enterprises' revenues.

Regarding the proxy variables, the minimum value of asset density ($AINT_{i,t-1}$) is 0.329, and the maximum value is 18.94, indicating significant variability in asset density among different enterprises. Similarly, the GDP growth rate ($\Delta GDP_{i,t-1}$) is highly variable across the different

**Table 2. Descriptive statistics of the main variables.**

| variable | N | Mean | P50 | Sd | Min | Max. |
|---|---|---|---|---|---|---|
| ROAVoli,t | 26198 | 0.031 | 0.019 | 0.0370 | 0 | 0.425 |
| Stickyi,t | 26198 | 0.176 | 0.0970 | 1.049 | -3.989 | 4.536 |
| AINTi,t-1 | 26198 | 2.421 | 1.854 | 2.031 | 0.329 | 18.94 |
| ΔGDPi, t-1 | 26198 | 10.26 | 9.647 | 6.592 | -9.972 | 69.81 |
| FCFi,t-1 | 26198 | 0.0480 | 0.0470 | 0.0690 | -0.223 | 0.283 |
| Shrzi,t | 26198 | 0.343 | 0.252 | 0.288 | 0.00500 | 1 |
| Comi,t | 26198 | 15.28 | 15.29 | 0.781 | 12.39 | 17.57 |
| Indratioi,t | 26198 | 37.48 | 36.36 | 5.389 | 25 | 60 |
| Sizei,t | 26198 | 22.32 | 22.13 | 1.297 | 19.32 | 26.45 |
| Levi,t | 26198 | 0.452 | 0.451 | 0.200 | 0.0270 | 0.908 |
| Growthi,t | 26198 | 0.165 | 0.101 | 0.420 | -0.658 | 4.024 |
| Agei,t | 26198 | 2.351 | 2.398 | 0.619 | 1.099 | 3.401 |
| Abslacki,t | 26198 | 0.241 | 0.123 | 13.16 | -0.00700 | 2126 |

cities where the listed enterprises are located, with certain cities experiencing negative growth in a specific year. The same is true for free cash flow ($FCF_{i,t-1}$), which is highly variable across enterprises, with some enterprises even experiencing negative free cash flow.

Regarding the control variables, the variability of equity concentration ($Shrz_{i,t}$) is significant, and some listed enterprises have high equity concentration, making it difficult for small and medium-sized shareholders to form adequate checks and balances on large shareholders. The mean value of growth ($Growth_{i,t}$) is 0.165, indicating that the operating income of listed enterprises has maintained an annual growth rate of 16.5% in recent years.

## 4.2 Industry distribution of cost stickiness

Since there exists the cost resources allocation variability among enterprises in different industries, observing this phenomenon helps to provide a more comprehensive understanding of this core variable.

First, this paper uses a t-test to determine the difference between high and low industrial competitiveness enterprises. In this paper, the industrial competitiveness is measured by the Herfindahl-Hirschman index (HHI index). Then, according to the median of the annual HHI index, the industries in which the enterprises are located are divided into two groups according to the degree of competitiveness. The HHI index is calculated as shown in Eq (6).

$$HHI_{i,t} = \sum \left( \frac{Sale_{i,t}}{Sale_{i,k,t}} \right)^2 \tag{6}$$

The HHI Index measures market concentration by summing the squares of the market shares of all firms in a market. A higher HHI indicates a more concentrated market. Where $Sale_{i,t}$ is the operating revenue of enterprise i in year t, and $Sale_{i,k,t}$ is the sum of the operating revenue of all enterprises in industry k in which enterprise i is located in year t. The larger the value of HHI, the higher the market concentration in the industry in the year, the higher the degree of monopolization in the industry, and the less intense the degree of competition.

Table 3 reports the t-test results of the cost stickiness indicator grouped by the degree of industry competitiveness. The mean value of cost stickiness ($Sticky_{i,t}$) is 0.18 in the low HHI group, which is larger than that of 0.155 in the high HHI group, and the t-test results also indicate that there is a difference in the means of the two samples at the 1 percent significance level. This result suggests that enterprises in highly competitive industries have higher cost stickiness. In more competitive industries, enterprises may have increased incentives to engage in horizontal mergers and acquisitions to achieve scale advantages, and enterprises may invest in more specialized assets that cannot be cut in time when the enterprise's scale decreases, resulting in high cost stickiness.

This paper provides further insight into the industry-specific distributional characteristics of cost stickiness by looking at the average cost stickiness of enterprises and the share of enterprises with cost stickiness in different industries.

**Table 3. T-test results of cost stickiness grouped by industry competitiveness.**

|  | Low HHI | | High HHI | | MeanDiff |
| --- | --- | --- | --- | --- | --- |
| Variable | N | Mean | N | Mean | |
| Stickyi, t | 11986 | 0.18 | 14014 | 0.155 | 0.041*** |

Note: LowHHI is the low HHI group, which indicates a higher degree of industry competition; HighHHI is the high HHI group, which indicates a lower degree of industry competition.

***, **, and * indicate significance at the 1%, 5%, and 10% levels, respectively.

**Table 4. Distribution of cost stickiness across industries.**

| Industry Code | Industry Name | Cost Stickiness Mean | Percentage of positive cost stickiness |
|:---:|:---:|:---:|:---:|
| A | Agriculture, forestry, animal husbandry and fisheries | 0.799 | 58.2% |
| B | Extractive industry | 0.176 | 55.8% |
| C | Service industry | 0.212 | 62.1% |
| D | Electricity, heat, gas, and water production and supply industry | 0.154 | 58.1% |
| E | Building industry | 0.116 | 60.5% |
| F | Wholesale and retail trade | 0.068 | 55.6% |
| G | Transportation, storage, and postal services | 0.089 | 56.8% |
| H | Accommodation and catering | 0.119 | 48.5% |
| I | Information transmission, software, and information technology services | 0.162 | 58.7% |
| K | Real estate industry | 0.014 | 55.5% |
| L | Leasing and business services | 0.186 | 62.5% |
| M | Scientific research and technical services | 0.171 | 64.6% |
| N | Water, environment, and utilities management | 0.167 | 58.4% |
| O | Residential services | 0.229 | 53.8% |
| P | Education | 0.218 | 52.1% |
| Q | Health and social work | -0.051 | 55.7% |
| R | Culture, sports, and recreation | 0.209 | 59.4% |
| S | Public administration, social security, and social organizations | 0.064 | 54.3% |

Table 4 reports the industrial distribution of cost stickiness. Regarding the proportion of positive cost stickiness in each industry, the number of enterprises characterized by cost stickiness in all industries except "Accommodation and catering" exceeds 50%, indicating that cost stickiness is prevalent in almost all industries. Regarding the mean value of cost stickiness within industries, the mean value of cost stickiness is positive in all industries except "Health and social work" industry. "Agriculture, forestry, animal husbandry, and fishery" industry has the highest cost stickiness value of 0.799, which may be attributed to the fact that the industry invests more in human capital, and enterprises will not cut costs by dismissing employees drastically when the short-term business volume declines. "Health and social work" has the lowest cost stickiness value at -0.051, which is particularly evident during the COVID-19 epidemic and the Post-Epidemic Era. Although human capital investment in this sector is high, it is a short-term investment. Once the public health event is over, there is a significant oversupply of human capital, leading enterprises in this sector to cut costs proportionately, or even more drastically, as business declines, and thus cost stickiness is generally low, or even anti-stickiness of costs is seen.

## 4.3 Main results regression results

**4.3.1 Correlation test.** Before conducting the multiple regression analysis of cost stickiness and business risk, the variables involved in the regression were first subjected to Pearson and Spearman correlation tests to clarify the correlation between the variables. The results of the correlation test are given in Table 5.

Table 5 reports the results of the correlation test. In this case, the Pearson correlation coefficient value is 0.028, and Spearman's correlation coefficient value is 0.032, and both are significant at the 1% level, indicating a significant correlation between the cost stickiness ($Sticky_{i,t}$) and enterprise risks ($ROAVol_{i,t}$). In addition, the control variables are all related to enterprise risks, indicating the necessity of year and industry control.

**Table 5. Correlation test results.**

| | ROAVol | Sticky | Shrz | Com | Indratio | Size | Lev | Growth | Age |
|---|---|---|---|---|---|---|---|---|---|
| ROAVol | | 0.032*** | 0.043*** | -0.117*** | 0.015** | -0.225*** | -0.119*** | -0.030*** | -0.059*** |
| Sticky | 0.028*** | | 0.022*** | 0.000 | 0.006 | -0.026*** | -0.035*** | -0.043*** | -0.064*** |
| Shrz | 0.055*** | 0.015** | | 0.137*** | 0.008 | -0.064*** | -0.102*** | 0.028*** | -0.153*** |
| Com | -0.079*** | 0.002 | 0.123*** | | -0.010 | 0.516*** | 0.089*** | 0.071*** | 0.103*** |
| Indratio | 0.018*** | 0.000 | -0.002 | -0.009 | | 0.001 | -0.028*** | -0.004 | -0.041*** |
| Size | -0.197*** | -0.018*** | -0.045*** | 0.532*** | 0.027*** | | 0.448*** | 0.064*** | 0.365*** |
| Lev | -0.059*** | -0.025*** | -0.094*** | 0.081*** | -0.024*** | 0.450*** | | 0.031*** | 0.284*** |
| Growth | 0.012* | -0.028*** | 0.015** | 0.012* | -0.000 | 0.051*** | 0.047*** | | -0.123*** |
| Age | -0.002* | -0.043*** | -0.131*** | 0.095*** | -0.036*** | 0.344*** | 0.292*** | -0.057*** | |

Lower-triangular cells report Pearson's correlation coefficients, and upper-triangular cells are Spearman's rank correlation

Note:

***, **, and * indicate significance at the 1%, 5%, and 10% levels, respectively.

**4.3.2 Multiple regression analysis.** Table 6 reports the results of the multiple regression of cost stickiness and enterprise risks.

In Table 6, the coefficient of cost stickiness ($Sticky_{i,t}$) in column (1) is positive and significant at the 1% level, proving the positive relationship between cost stickiness and enterprise

**Table 6. Multiple regression results of cost stickiness and enterprise risks.**

| Dep. Var | $ROAVol_{i,t}$ | | |
|---|---|---|---|
| | **(1)** | **(2)** | **(3)** |
| $Sticky_{i,t}$ | 0.001*** | 0.001*** | 0.001** |
| | (3.01) | (3.07) | (2.45) |
| $Shrz_{i,t}$ | | 0.007*** | 0.006*** |
| | | (8.54) | (8.38) |
| $Com_{i,t}$ | | 0.002*** | 0.000 |
| | | (5.76) | (1.07) |
| $Indratio_{i,t}$ | | 0.000*** | 0.000*** |
| | | (4.87) | (3.24) |
| $Size_{i,t}$ | | -0.008*** | -0.007*** |
| | | (-26.67) | (-25.18) |
| $Lev_{i,t}$ | | 0.007*** | 0.012*** |
| | | (4.36) | (7.00) |
| $Growth_{i,t}$ | | 0.002*** | 0.003*** |
| | | (3.34) | (3.77) |
| $Age_{i,t}$ | | 0.005*** | 0.005*** |
| | | (12.95) | (13.04) |
| _cons | 0.031*** | 0.143*** | 0.165*** |
| | (129.08) | (28.46) | (27.28) |
| Year | No | No | Yes |
| Industry | No | No | Yes |
| N | 26251 | 26198 | 26198 |
| Adj. R$^2$ | 0.601 | 0.651 | 0.689 |

Note:

***, **, and * indicate significance at the 1%, 5%, and 10% levels, respectively.

risks. In columns (2) and (3), the coefficient of cost stickiness (Sticky$_{i,t}$) is still significant at the 1% level after the gradual addition of control variables as well as year and industry effects, and the adjusted $R^2$ of the results in column (3) continues to improve after controlling for industry and year effects, which illustrates the necessity of controlling for the control variables as well as year and industry effects. These findings underscore the importance of incorporating cost stickiness into risk management strategies and highlight the need for further research to unpack the mechanisms through which cost stickiness influences enterprise risks across different industry and economic contexts.

The results of Tables 3–6 validate Hypothesis H1: Cost stickiness is positively related to enterprise risk, and enterprises in different industries have different cost stickiness. This result is consistent with the findings of Subramanian and Weidenmier (2003) [61], indicating that the clustering of enterprises and their varying competitiveness are indeed associated with cost stickiness.

## 4.4 Endogeneity tests

In this section, we make endogeneity tests with the help of instrumental variables, and consistent estimations are obtained by regressing using the portion of the endogenous variable that is uncorrelated with the disturbance term. This paper uses the median cost stickiness of the same province in the same year (MSticky$_{i,t}$) as an instrumental variable for cost stickiness for the following reasons. First, cost stickiness at the regional level does not directly affect enterprise risk at the enterprise level, i.e., it satisfies the exogeneity requirement. Second, since the cost stickiness at the regional level is computed from the cost stickiness at the enterprise level, a certain degree of numerical correlation exists between the two variables, and there are no weak instrumental variables.

Using the instrumental variables approach presupposes the existence of endogenous explanatory variables, and this paper takes two approaches to test this. First, a Hausman test is conducted with the original hypothesis that all explanatory variables are exogenous, i.e., no endogenous variables exist. Second, a heteroskedasticity robust DWH test is conducted to compensate for the Hausman test not holding in the heteroskedasticity case.

From the results of Table 7, the p-value of both the Hausman and DWH tests is less than 10%, rejecting the original hypothesis that the cost stickiness variable is exogenous and can be considered an endogenous explanatory variable. The recognition of cost stickiness as an endogenous variable allows for a more nuanced understanding of its dynamic relationship with enterprise risks, potentially leading to more effective risk management strategies.

Table 8 reports the 2SLS regression results. Columns (1) and (2) show the results of the first stage and second stage regressions, respectively. The first stage regression is a regression of instrumental variables on cost stickiness, and the second stage regression is a regression of enterprise risks on the exogenous component of cost stickiness. The three statistics results at the bottom of the table are the unidentifiable test, the fit of the first-stage regression model, and the weak instrumental variable test. In column (2), the coefficient value on cost stickiness (Sticky$_{i,t}$) rises to 0.004 and it is significant at the 10% level, validating the robustness of the regression results. The significant coefficient increase in cost stickiness in the second-stage

**Table 7. Endogeneity test results.**

|  | Hausman | Durbin | Wu-Hausman |
|---|---|---|---|
| chi2/F | 2.04 | 2.03 | 2.04 |
| *p* | 0.0536 | 0.0527 | 0.053 |

regression, along with the supportive statistical results, reinforce the conclusion that cost stickiness is an endogenous factor that significantly influences enterprise risks.

To further examine the correlation between the instrumental and endogenous variables, its unidentifiable test shows that the Kleibergen-Paaprk LM statistic is 159.227 with a p-value of 0.0000, so the original hypothesis of unidentifiable is rejected. From the results of the one-stage regression in column (1), it can be seen that the instrumental variables have good explanatory power for the endogenous variables, and the coefficients are significantly positive at the 1% level. Meanwhile, the F-statistic of the one-stage regression is 207.598 with a p-value of 0.0000, which rejects the original hypothesis that the coefficient of the instrumental variable is 0 in the first-stage regression and further verifies the correlation between instrumental variables and endogenous variables.

## 4.5 Robustness tests

In this section, we enhance the credibility of our findings by conducting robustness tests through the substitution of enterprise risk indicators and the alteration of the regression sample. This approach is a standard procedure in empirical research to ensure that the results are not contingent on a specific measurement or sample selection. By employing alternative indicators, we can ascertain whether the observed relationship between cost stickiness and enterprise risks is consistent. Additionally, the sample replacement allows us to evaluate the generalizability of our findings, confirming that they are not an artifact of the initial sample composition. These tests are crucial for establishing the robustness of our conclusions and for providing confidence in the validity of our economic models.

### 4.5.1 Replacement of enterprise risks indicators.

1. Referring to Faccio et al. [69], we use an industry-adjusted return on equity volatility (ROEVol$_{i,t}$) as a proxy for enterprise risks. The standard deviation of ROEVol reflects both the risk of enterprise projects and the additional risk arising from the use of leverage in the capital structure.

2. Referring to He et al. [70], we calculated the volatility of the net rate of return on assets (ROAVol2$_{i,t}$) as the standard deviation of the industry-adjusted return on assets from year t to year t+2. We replaced the indicator of enterprise risks with it.

**Table 8. 2SLS regression results.**

| Dep. Var | $Sticky_{i,t}$ | $ROAVol_{i,t}$ |
|---|---|---|
| | (1) | (2) |
| MSticky$_{i,t}$ | 1.132*** | |
| | (14.41) | |
| Sticky | | 0.004* |
| | | (1.80) |
| Controls | Yes | Yes |
| Year | Yes | Yes |
| Industry | Yes | Yes |
| *Kleibergen-Paap rk LM statistic* | | 159.227*** |
| *First-stage F statistic* | | 207.598*** |
| *Minimum eigenvalue statistic* | | 246.97 |
| N | 26198 | 26198 |
| Adj.$^2$ | 0.520 | 0.582 |

Note:

***, **, and * indicate significance at the 1%, 5%, and 10% levels, respectively.

3. Referring to Ohlson [71], we calculated the risk coefficient by O-Score (Oscore Risk$_{i,t}$) and replaced the indicator of enterprise risks with it. The O-Score is a financial metric used to predict the probability of an enterprise's bankruptcy. A higher O-score indicates a greater risk of bankruptcy. The Oscore Risk$_{i,t}$ indicator is calculated through a series of financial indicators, which can better reflect the enterprise's bankruptcy probability. The higher the value, the higher the risk of the enterprise. The calculation method is shown in Formula (7) and Formula (8).

$$\text{Oscore}_{i,t} = -1.32 - 0.407 \times \text{Size}_{i,t} + 6.03 \times \text{Lev}_{i,t} - 1.43 \times \text{WCTA}_{i,t}$$
$$+ 0.0757 \times \text{CLCA}_{i,t} - 2.37 \times \text{NITA}_{i,t} - 1.83 \times \text{FUTL}_{i,t} + 0.285 \times \text{INTWO}_{i,t} \quad (7)$$
$$- 1.72 \times \text{OENEG}_{i,t} - 0.521 \times \text{CHIN}_{i,t}$$

$$\text{OscoreRisk}_{i,t} = \frac{e^{\text{Oscore}_{i,t}}}{1 + e^{\text{Oscore}_{i,t}}} \times 100 \quad (8)$$

Where Size$_{i,t}$ is the total assets taken in logarithms, Lev$_{i,t}$ is the total liabilities/total assets, WCTA$_{i,t}$ is the working capital/total assets at the end of the period, CLCA$_{i,t}$ is the current liabilities/current assets, NITA$_{i,t}$ is the net profit/total assets, and FUTL$_{i,t}$ is the net cash flow from operations/total liabilities.

INTWO$_{i,t}$ is a dummy variable for the last two years when net profit was negative, OENEG$_{i,t}$ is a dummy variable for Lev1$_{i,t}$, and CHIN$_{i,t}$ is the rate of change in net profit (the denominator is taken as the sum of the absolute values of the last two years' net profit).

Table 9 reports the results of the robustness tests with the replacement of the enterprise risks indicator, and the coefficients on cost stickiness (Sticky$_{i,t}$) are all significantly positive at the 1% level after replacing the enterprise risk indicator, indicating the robustness of the regression results. The positive relationship between cost stickiness and enterprise risks is thus confirmed to be a robust empirical regularity, suggesting that as cost stickiness increases, so does the level of enterprise risks. This outcome highlights the importance of considering cost stickiness in the broader context of risk management and strategic decision-making within enterprises.

**4.5.2 Replacement of regression samples.**

1. Take the manufacturing industry as a sub-sample. Weiss [56] pointed out that the homogeneity of profit and cost components of manufacturing enterprises allows for a more

**Table 9. Robustness test results after replacing enterprise risk indicators.**

| Dep.Var | ROEVol$_{i,t}$ | ROAVOL2$_{i,t}$ | OscoreRisk$_{i,t}$ |
|---|---|---|---|
|  | (1) | (2) | (3) |
| *Sticky$_{i,t}$* | 0.537*** | 0.261*** | 0.057 |
|  | (5.71) | (6.65) | (5.11) |
| Controls | Yes | Yes | Yes |
| Year | Yes | Yes | Yes |
| Industry | Yes | Yes | Yes |
| N | 26198 | 26198 | 26198 |
| Adj. R$^2$ | 0.543 | 0.540 | 0.544 |

Note: The explanatory variables in columns (1) through (3) are return on net assets volatility, return on assets volatility over a three-year window, and risk coefficients calculated using the O-Score, respectively.

***, ** and * indicate significance at the 1%, 5% and 10% levels, respectively.

accurate analysis of the impact of cost sticky behavior on profit surplus. Also, since manufacturing enterprises are in a fully competitive market, modeling the operating revenue of manufacturing enterprises as their volume measurement can reduce the measurement bias caused by product pricing fluctuations.

2. Take enterprises with positive cost stickiness as a sub-sample. Due to cost anti-stickiness [3], enterprises will cut costs significantly when operating revenues fall. Robustness tests using this sub-sample with positive cost stickiness can further clarify the relationship between cost stickiness and enterprise risks.

3. Excluding samples whose cost rate fluctuations are in the top 10% of the industry. Since the cost stickiness index measured by the Weiss model uses quarterly data with large seasonal fluctuations, excluding samples with high fluctuations in cost rates during the year can effectively mitigate the impact of seasonal fluctuations in business on the results. This paper expresses the fluctuations in the total cost rate of the enterprise in the year in terms of its standard deviation.

As seen in Table 10, after replacing the regression sample, the main effect remains significantly positive at the 1% level, indicating that the relationship between cost stickiness and the positive association with enterprise risks is robust. This robustness adds credibility to the economic models that incorporate cost stickiness as a key variable in risk assessment, indicating that the relationship is not an artifact of the sample selection but a substantive economic phenomenon.

## 4.6 Mediated effects test

This paper examines the mediating effect according to Eqs (3) through (5). It involves three steps: first, establishing the effect of the independent variable on the mediator; second, establishing the effect of the mediator on the dependent variable; and third, examining the direct effect of the independent variable on the dependent variable while controlling for the mediator. The mediated effects test helps to determine whether the mediator accounts for the entire effect, or just a part of it, providing a more nuanced understanding of causal pathways.

Table 11 reports the the test results of the mediating role of absorbed slack. Column (1) tests whether cost stickiness increases enterprise risks. The coefficient value of cost stickiness (Sticky$_{i,t}$) is 0.001 and is significant at the 1% level, which aligns with the multivariate regression results in Table 6. In column (2), with absorbed slack as the explanatory variable and cost

**Table 10. Robustness test results for the replacement regression sample.**

| Dep. Var | $ROAVol_{i,t}$ | | |
|---|---|---|---|
| | **Ind = C** | **Sticky>0** | **Non-10% VOL** |
| | **(1)** | **(2)** | **(3)** |
| *Sticky$_{i,t}$* | 0.203*** | 0.493*** | 0.231*** |
| | (4.86) | (8.57) | (5.95) |
| Controls | Yes | Yes | Yes |
| Year | Yes | Yes | Yes |
| Industry | Yes | Yes | Yes |
| N | 16301 | 15833 | 22844 |
| Adj. R$^2$ | 0.541 | 0.583 | 0.652 |

Note: The samples in columns (1) through (3) are the manufacturing sub-sample, the sub-sample with positive cost stickiness, and the sub-sample that excludes the top 10% of cost rate fluctuations, respectively. Each regression controls for industry and year effects.

***, **, and * denote significance at the 1%, 5%, and 10% levels, respectively.

**Table 11. Results of mediation effect tests for absorbed slack.**

| Dep. Var | $RoaVol_{i,t}$ | $Abslack_{i,t}$ | $RoaVol_{i,t}$ |
|---|---|---|---|
| | (1) | (2) | (3) |
| $Sticky_{i,t}$ | 0.001*** | 0.005*** | 0.001*** |
| | (3.74) | (6.09) | (3.43) |
| $Abslack_{i,t}$ | | | 0.022*** |
| | | | (8.10) |
| Controls | Yes | Yes | Yes |
| Year | Yes | Yes | Yes |
| Industry | Yes | Yes | Yes |
| Sobel Z | | 6.407*** | |
| Proportion that is mediated | | 15.76% | |
| N | 20952 | 20952 | 20952 |
| Adj. R$^2$ | 0.590 | 0.624 | 0.595 |

Note: Columns (1) through (3) of the table show the regressions for mediated effects path c, path a, and paths b & c',
respectively, and the two statistics at the bottom of the table are the Sobel test and the mediated effect as a proportion
of the total effect, respectively. Each regression controls for industry and year effects.

***, ** and * indicate significance at the 1%, 5% and 10% levels, respectively.

stickiness as the explanatory variable, the regression coefficient value of cost stickiness ($Sticky_{i,t}$) is 0.005 and significant at 1% level, indicating that cost stickiness is significant and positively related to absorbed slack. Column (3) is formed after adding absorbed slack to column (1) to verify whether the mediating role of absorbed redundancy holds. In column (3), the coefficient value of cost stickiness ($Sticky_{i,t}$) and the coefficient b of absorbed slack ($Abslack_{i,t}$) are 0.001 and 0.022, respectively. Both pass the significance test at the 1% level and Sobel's test, proving that absorbed slack is a mediator of the impact of cost stickiness on enterprise risk. Its mediating effect accounts for 15.76% of the total effect. The consistent statistical significance of the coefficients at the 1% level across different model specifications not only reinforces the theoretical underpinnings of the hypothesis but also offers empirical evidence of the nuanced mechanisms through which cost stickiness influences enterprise risks. The above results verify hypothesis H2 that cost stickiness increases enterprise risks by increasing the absorbed redundancy of enterprises. This outcome aligns with the research findings of Love E G and Nohria N (2005) [62], as well as Ball et al. (2022) [66], indicating that absorbable slack plays a mediating role in the relationship between cost stickiness and enterprise risks.

At the same time, the mediating mechanism of "cost stickiness-absorbed slack-enterprise risks" is tested again using the bootstrap method with 95% confidence intervals, and the sample size is set at 1,000. According to Wen and Ye [72], when the 95% confidence interval does not contain zero, the mediation effect is determined to be significant.

As seen in Table 12, the 95% confidence intervals do not contain 0 in both the indirect and direct effects, so it can be assumed that both the indirect effect of absorbed slack and the direct effect of cost stickiness on firm risk hold significantly. This statistical significance indicates that the mediating role of absorbed slack is not only present but also a substantial component of the overall effect, reinforcing the theoretical and practical implications of hypothesis H2.

## 5. Discussion

In the realm of empirical research, this study delves into the intrinsic relationships between cost stickiness, absorbed slack, and enterprise risks, yielding several pivotal conclusions.

**Table 12. Bootstrap test results.**

|  | Observed eff. | Bootstrap Std. Err. | Z | Normal-based |
|---|---|---|---|---|
|  |  |  |  | [95% Conf. Interval] |
| Indirect effect | 0.0001141 | 0.0000242 | 4.72*** | 0.0001615 |
| Direct effect | 0.0011664 | 0.0003358 | 3.47*** | 0.0018245 |

Note: Indirect effect is the indirect effect, and Direct effect is the direct effect. Observed eff. is the coefficient obtained from the bootstrap method.

***, ** and * denote significance at the 1%, 5% and 10% levels, respectively.

Firstly, the study affirms the ubiquity of cost stickiness, which is prevalent across most industries with significant variances attributed to industry-specific characteristics. This revelation suggests that cost stickiness, as a pervasive phenomenon, may manifest differently within various industry contexts due to distinct operational patterns and cost structures.

Secondly, the research establishes a robust positive correlation between cost stickiness and enterprise risks. Specifically, enterprises with higher degrees of cost stickiness face greater operational risks, potentially because cost stickiness impedes flexibility in the face of market volatility and hinders the ability to promptly adjust cost structures in response to external environmental changes. Moreover, the presence of cost stickiness may obscure the true financial health of an enterprise, thereby affecting managerial decisions and increasing financial risks.

Lastly, the study reveals that cost stickiness augments enterprise risks by increasing the level of absorbed slack within enterprises. This finding underscores the critical need for a sophisticated approach to cost management, as the inflexibility in cost structures can lead to the accumulation of idle resources, or absorbed slack, exacerbating the risks confronted by an organization. This relationship highlights the importance of understanding and managing cost stickiness to mitigate risks and enhance organizational resilience in strategic cost management and organizational resilience.

## 6. Conclusion, implication and limitation

### 6.1 Conclusion

The interplay between cost stickiness, absorbed slack, and enterprise risks is a critical area of management research, particularly for Chinese enterprises where it has become a central element of their sustainable development strategies. This study provides an in-depth examination of the nexus among these three constructs.

Empirical analysis within this research has substantiated the pervasive presence of cost stickiness across a spectrum of enterprises, reflecting the asymmetrical adjustment of costs within organizational operations and posing challenges to financial management and strategic planning. The study also delineates the distributional characteristics of cost stickiness across various industries, offering a more granular perspective on the behavioral patterns of cost stickiness under different economic conditions.

Furthermore, the research establishes a strong positive correlation between cost stickiness and both absorbed slack and enterprise risks. The economic ramifications of cost stickiness are explored by introducing the variable of absorbed slack to assess its mediating role in the relationship between cost stickiness and enterprise risks. Empirical findings indicate that cost stickiness exerts a partial mediating effect on enterprise risks through absorbed slack, augmenting the understanding of the impact of absorbed slack on cost stickiness.

Additionally, this paper integrates cost stickiness, absorbed slack, and enterprise risks within a cohesive theoretical framework for analysis. Under this framework, the study further

validates the robustness of the findings by substituting enterprise risk indicators and research samples, thereby broadening the scope and outcomes of research related to cost stickiness.

In conclusion, cost stickiness has emerged as a significant impediment to the achievement of sustainable development by enterprises. The findings of this study not only offer a novel perspective on the complex interplay between cost stickiness, absorbed slack, and enterprise risks but also provide a solid theoretical foundation and practical guidance for mitigating the adverse impact of cost stickiness on enterprise risks and leveraging the mediating role of absorbed slack to enterprise risks. These insights underscore the potential value of absorbed slack in enhancing enterprise adaptability and reducing risks, contributing new insights to academic research and practical operations in related domains.

## 6.2 Suggestion and implication

Addressing cost stickiness, absorbed slack and enterprise risks requires a proactive and strategic approach that encompasses cost management, risk assessment, operational agility, and the use of technology. Enterprises can enhance their resilience and competitiveness in the face of market uncertainties and operational challenges. It is essential for them to continuously monitor and adapt their strategies to stay ahead of potential risks and capitalize on emerging opportunities. Based on the study's findings, this paper proposes the following suggestions.

(1) Enterprises should pay more attention to the phenomenon of cost stickiness and recognize the impact of cost stickiness on their operations. First, they should make full risk assessment and Management. A robust risk assessment process is essential for identifying potential enterprise risks. This process should involve a comprehensive evaluation of internal and external factors that could impact the enterprise. Once risks are identified, a proactive risk management strategy should be developed. This includes establishing risk mitigation plans, contingency plans, and insurance coverage where appropriate. In resource allocation, operating profit, short-term decision-making, and the formulation of cost management strategies, cost stickiness and its impact on enterprise risks should be fully considered. Enterprises should strengthen risk prevention, improve risk management awareness, pay attention to changes in cost stickiness, and improve risk management awareness. Second, they should adopt agile business practices. Agile business practices enable enterprises to respond quickly to changes in the market and internal operations. This approach involves breaking down projects into smaller, manageable tasks and regularly reassessing priorities based on current conditions. By adopting agility, enterprises can better manage absorbed slack by quickly adjusting production levels and resource allocation to meet demand changes. Third, they should implement advanced cost management techniques.

Advanced cost management techniques, such as activity-based costing and value chain analysis, can provide deeper insights into cost behavior. These methods help in identifying cost drivers and understanding how costs are affected by changes in business activities. By leveraging these techniques, enterprises can make informed decisions about where to allocate resources and how to adjust operations in response to market fluctuations. In daily business activities, enterprises should strengthen the control and prediction of costs to avoid expanding or reducing the scale of business; establish a sound risk management system and internal control mechanism to reduce the impact of cost stickiness on enterprise risks; establish a cost management system, accurately grasp all cost information, discover the causes of cost fluctuations on time, and continuously optimize the production process and reduce the cost of raw materials; establish a perfect risk management system, and regularly identify and evaluate market risks, supply chain risks, legal and regulatory risks, and take timely countermeasures.

(2) Enterprises should strengthen the management of absorbed slack. Enterprises should reduce the impact of absorbed slack on enterprise risks by optimizing resource allocation, improving operational efficiency, reducing resource waste; reduce the level of cost stickiness by strengthening internal management, improving production efficiency, and reducing inventory, and at the same time, rationally arrange the capital structure to reduce financial risk; improve the comprehensive quality and decision-making level of the management team by strengthening training, selecting excellent talents, and establishing incentive mechanisms. First, they should diversify operations. Diversifying operations can help spread risk and reduce the impact of absorbed slack. By expanding into new markets or product lines, enterprises can create additional revenue streams that are not as susceptible to the same market fluctuations. This diversification can also provide opportunities to leverage synergies between different enterprise segments, leading to cost efficiencies. Second, they should leveragie technology and automation.

The use of technology and automation can help reduce cost stickiness by increasing operational efficiency. Automated systems can quickly adjust to changes in production levels, reducing the need for excess capacity. Additionally, technology can provide real-time data that helps in making informed decisions about resource allocation and cost management. At the mean time, they should enhance supply chain resilience. A resilient supply chain is less likely to be disrupted by absorbed slack and enterprise risks. Enterprises should invest in supply chain management systems that provide visibility and flexibility. This includes developing strong relationships with suppliers, implementing just-in-time inventory practices, and having backup suppliers in place to ensure continuity of operations. Third, they should invest employee Training and Development. Investing in employee training and development can help organizations adapt to changes more effectively. Employees who are well-trained and have a broad skill set are better equipped to handle fluctuations in workload and can contribute to cost management efforts. This includes training in areas such as lean manufacturing, process improvement, and financial analysis.

(3) Enterprises should strengthen information disclosure and improve the quality of accounting information. Enterprises should establish and improve the internal control mechanism and financial management system, standardize the information disclosure process and content, and ensure the accuracy and completeness of financial reports. First, they should make financial planning and forecasting. Accurate financial planning and forecasting are crucial for managing absorbed slack and enterprise risks. Enterprises should develop robust financial models that account for various scenarios and uncertainties. This includes stress testing financial plans to ensure that the organization can withstand potential shocks to its operations. Second, they should cultivate the culture of openness, honesty and transparency, incorporate it into the values and code of conduct of the enterprise, and enhance the awareness and sense of responsibility of the internal staff for information disclosure. Enterprises should strengthen the internal auditing, improve the external regulatory system and improve the quality of information disclosure, and disclose the information of financial statements, financial analysis, changes in significant accounting policies in accordance with the requirements of the relevant regulations and accounting standards. Also, they should disclose information such as financial statements, financial analysis, significant accounting policy changes, to improve the comprehensiveness and consistency of information, reduce the degree of information asymmetry. Third, they should maintain regulatory compliance and ethical practices. Maintaining regulatory compliance and adhering to ethical practices can help mitigate risks. Enterprises should ensure that they have systems in place to monitor and comply with relevant laws and regulations. Additionally, fostering a culture of ethics

and transparency can help build trust with stakeholders and reduce the likelihood of risks related to non-compliance.

## 6.3 Research limitation and recommendation

In the exploration of the dynamics between cost stickiness, absorbed slack, and enterprise risks, this study acknowledges certain limitations that suggest avenues for future inquiry.

Firstly, the research is circumscribed to Chinese enterprises, without extending to businesses in other nations. Given China's position as the largest developing country and an emerging economy, the study of its enterprises holds a certain level of representativeness. Recognizing that cost stickiness is a ubiquitous phenomenon across businesses, future studies could expand their scope to encompass enterprises from various countries. This expansion would facilitate an evaluation of how cost stickiness influences absorbed slack and risk management in diverse settings, allowing for comparative analyses with the outcomes of this research.

Secondly, the temporal scope of the sample used in this study is relatively narrow. Utilizing a dataset spanning from 2007 to 2022, this research has established the pervasiveness of cost stickiness, the critical role of absorbed slack, and the imperative of risk management. However, whether these phenomena will persist as long-term trends remains to be seen. Consequently, future studies could extend the timeframe of their samples to ascertain the longevity of these conclusions over time.

## Author Contributions

**Conceptualization:** Qian Binhua, Yang Boyuan.

**Data curation:** Qian Binhua, Yang Boyuan.

**Formal analysis:** Qian Binhua.

**Funding acquisition:** Qian Binhua.

**Investigation:** Qian Binhua.

**Methodology:** Qian Binhua.

**Project administration:** Qian Binhua.

**Resources:** Qian Binhua.

**Software:** Qian Binhua.

**Supervision:** Qian Binhua.

**Validation:** Qian Binhua.

**Visualization:** Qian Binhua.

**Writing – original draft:** Qian Binhua.

**Writing – review & editing:** Qian Binhua.

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
