## [Decision Letter · Decision Letter 0]

31 Oct 2024

PONE-D-24-28797Cost Stickiness, Absorbed Slack and Enterprise Risks: Evidence from ChinaPLOS ONE

Dear Dr.Qian,

Thank you for submitting your manuscript to PLOS ONE. After careful consideration, we feel that it has merit but does not fully meet PLOS ONE’s publication criteria as it currently stands. Therefore, we invite you to submit a revised version of the manuscript that addresses the points raised during the review process.

We look forward to receiving your revised manuscript.

Kind regards,

Chen-Wei Yang

Academic Editor

PLOS ONE

Journal Requirements:

“This paper is supported by National Social Science Foundation of China Key Project "Research on Group Evaluation Consensus Mechanism Based on Trust Relationship" (22ATJ003) , National Key Project of Statistical Research "Comprehensive Evaluation Technology and Intelligent Application Based on Social Network" (2021LZ33), and Ningbo Jinsui Tax Big Data Research Base "Research on Strategies to Promote Research and Development of Technological SMEs" .

We would like to thank the anonymous reviewers for their helpful remarks.”

Additional Editor Comments (if provided):

The manuscript titled "Cost Stickiness, Absorbed Slack, and Enterprise Risks: Evidence from China" presents a solid and well-structured study based on an extensive dataset of Chinese enterprises. Both reviewers find the research to be of value, but they offer constructive feedback on improving clarity, tone, and depth of discussion, as well as expanding on theoretical implications. We look forward to receiving a revised version that reflects these changes.

Reviewers' comments:

Reviewer's Responses to Questions

**Comments to the Author**

1. Is the manuscript technically sound, and do the data support the conclusions?

Reviewer #1: Yes

Reviewer #2: Yes

2. Has the statistical analysis been performed appropriately and rigorously? 

Reviewer #1: Yes

Reviewer #2: Yes

3. Have the authors made all data underlying the findings in their manuscript fully available?

Reviewer #1: Yes

Reviewer #2: Yes

4. Is the manuscript presented in an intelligible fashion and written in standard English?

Reviewer #1: Yes

Reviewer #2: Yes

5. Review Comments to the Author

Reviewer #1: Dear Authors,

• Use a more academic tone in the paper. For example, the abstract is full of “intriguing”, “enigmatic”, “captivated”, “delicately” etc. The same idea applies to the first paragraphs of the paper. It is not a novel, but a research paper and the tone should be more objective. It is not pleasant to read.

• Some acronyms should be presented the first place they are mentioned (like HHI index).

• For the results part - the comparison between enterprises, considering the industrial competitiveness is first introduced at page 13, and not as a premise for the hypotheses. I suggest that the fact that the enterprises are clustered should be part of the hypotheses’ development, thus changing them in order to incorporate this high/low competitiveness.

• The multitude of variables used in the models makes the first part of the paper, although interesting, a little too simplistic.

• Also, regarding the variables. The reference no. 65 is considered representative for the control variables. However, I cannot find it on Scholar or on Google (generic search). I recommend you find another paper which should complement this one, the relevance of the first one being a bit under question.

• When presenting the results, other papers which sustain (or not) your findings should be part of the discussions.

• The discussion part is missing.

• The paper is interesting and has a lot of potential.

Reviewer #2: The manuscript is technically sound. The research is well-structured and based on a large dataset of Chinese enterprises. The findings are rigorously tested using various empirical methods, including robustness tests and endogeneity checks. More comments:

- A more detail on the theoretical implications of the results would strengthen the manuscript.

- A clearer explanation of some of the statistical techniques used (e.g., interaction effects) would improve clarity.

- The statistical methods used are appropriate, but the explanation of robustness checks and endogeneity tests could be expanded for clearer understanding.

- Moderate language editing is required to simplify sentence structure and ensure clarity.

- Suggestions for future research are lacking.

6. PLOS authors have the option to publish the peer review history of their article (what does this mean?). If published, this will include your full peer review and any attached files.

Reviewer #1: No

Reviewer #2: No

---

## [Author Response · Author response to Decision Letter 0]

5 Nov 2024

Dear Chen-Wei,

We are grateful for the opportunity to submit a revised version of our manuscript and appreciate the thorough and constructive feedback provided by you and the reviewers. We are now submitting the revised manuscript titled “Cost Stickiness, Absorbed Slack and Enterprise Risks: Evidence from China” for consideration for publication in PLOS ONE.

In our revision, we have meticulously addressed all the suggestions and comments made by the reviewers and editor. The key changes and additions are as follows:

1. We have made careful revisions to each point raised by the reviewers, with all changes tracked and visible in the manuscript.

2.In accordance with the PLOS ONE style template, we have formatted the entire manuscript to ensure compliance with the formatting requirements of PLOS ONE. 

3. We have issued a revised statement declaring all sources of funding or support received during the course of this study, and have included a statement in the updated funding declaration that "this research did not receive any additional external funding."

4. In compliance with the submission requirements, we have strictly adhered to the open data policy of PLOS ONE.

5. We have meticulously reviewed the reference list to ensure its completeness and accuracy, and to conform to the reference citation requirements of PLOS ONE.

We are confident that this revised manuscript has addressed your concerns. Given the esteemed reputation of your journal, we earnestly hope that our paper may be granted the honor of publication within your esteemed pages. We eagerly await your feedback.

Sincerely,

Qian Binhua 

Zhejiang business technology institute, Ningbo, Zhejiang Province, China

Enclosed: Responses to the comments from Reviewer 1 and 2.

Rebuttal Letter

Reply to Reviewer #1

Dear Reviewer,

We are grateful for the time you have dedicated to reviewing our manuscript and for your encouraging comments on its merits. We hope the following explanations and revisions have adequately addressed your concerns. In the remainder of this letter, we will discuss your comments individually and provide our corresponding responses.

To facilitate this discussion, we will restate your comments in italic font and then present our responses to them.

Comment 1:

Use a more academic tone in the paper. For example, the abstract is full of “intriguing”, “enigmatic”, “captivated”, “delicately” etc. The same idea applies to the first paragraphs of the paper. It is not a novel, but a research paper and the tone should be more objective. It is not pleasant to read.

Response 1:

Your suggestion is highly relevant. In response, during the revision process of the manuscript, we have focused on refining the tone of the writing to express it in a more academic manner. We have rewritten the entire abstract, eliminating any exaggerated expressions and adopting an objective tone for communication. Similarly, the first paragraph of the paper has also been rewritten in an objective tone, aiming to provide a better reading experience for the readers.

Comment 2:

Some acronyms should be presented the first place they are mentioned (like HHI index).

Response 2:

Thank you for the detailed references. In response to this comment, we have conducted a review of the acronyms used within the paper. For expressions such as the Herfindahl-Hirschman Index (HHI), we have provided a comprehensive definition at their first occurrence.

Comment 3:

For the results part - the comparison between enterprises, considering the industrial competitiveness is first introduced at page 13, and not as a premise for the hypotheses. I suggest that the fact that the enterprises are clustered should be part of the hypotheses’ development, thus changing them in order to incorporate this high/low competitiveness.

Response 3:

Your suggestions have been very instructive. We have included enterprise clustering as part of our hypothesis, adding a fourth point to the first hypothesis in Section 2.1: "Enterprises exhibit a tendency to cluster, which in turn influences the degree of cost stickiness." We have also refined the first hypothesis to: "H1: Cost stickiness is positively related to enterprise risk, and enterprises in different industries have varying levels of cost stickiness."

Comment 4:

The multitude of variables used in the models makes the first part of the paper, although interesting, a little too simplistic.

Response 4:

As you pointed out, the first part was somewhat simplistic. In response, we have enhanced the first part of the paper, providing a clearer definition of enterprise risks and further elaborating on the relationship between cost stickiness, absorbed slack, and enterprise risks. Additionally, we have included a more detailed discussion on how cost stickiness affects enterprise risks, thereby enriching the content of the first section.

Comment 5:

Also, regarding the variables. The reference no. 65 is considered representative for the control variables. However, I cannot find it on Scholar or on Google (generic search). I recommend you find another paper which should complement this one, the relevance of the first one being a bit under question.

Response 5:

Thank you for your review. We have identified an additional reference to replace the original reference number 65. The new reference is the paper titled "Using Accounting Earnings and Aggregate Economic Indicators to Estimate Firm-Level Systematic Risk," authored by Ball et al. and published in 2022.

Comment 6:

When presenting the results, other papers which sustain (or not) your findings should be part of the discussions.

Response 6:

Thank you for pointing out this issue. In the revised version, we have included other papers that either support or do not support our findings as part of the discussion, primarily in Section 4.

Comment 7:

The discussion part is missing.

Response 7:

Thank you for your careful review. We have added a Discussion section, as detailed in Section 5.

Comment 8:

The paper is interesting and has a lot of potential.

Response 8:

We appreciate your encouragement and assistance, and are pleased that our paper has received your approval.

In summary, we would like to thank you for all your time involved and for this excellent opportunity for us to improve the manuscript. We hope you will find this revised version satisfactory.

Sincerely,

Qian Binhua

Zhejiang business technology institute, Ningbo, Zhejiang Province, China

Rebuttal Letter

Reply to Reviewer #2

Dear Reviewer,

Thank you very much for your time in reviewing the manuscript and for your encouraging comments on its merits. We also appreciate your clear and detailed feedback and hope the explanation has fully addressed your concerns. In the remainder of this letter, we discuss your comments individually and provide our corresponding responses.

To facilitate this discussion, we first retype your comments in italic font and then present our responses to the comments.

Comment 1:

A more detail on the theoretical implications of the results would strengthen the manuscript.

Response 1:

We strongly agree with your point. We have provided a more detailed explanation of the theoretical implications of our findings, which can be found in Section 4.

Comment 2:

A clearer explanation of some of the statistical techniques used (e.g., interaction effects) would improve clarity.

Response 2:

We wholeheartedly agree with your opinion. In the revised manuscript, we have provided a more detailed explanation of the statistical methods used, including an elucidation of the three-step approach to mediating effects, which can be found in Subsection 4.6 ,in order to enhance the clarity and readability of the paper.

Comment 3:

The statistical methods used are appropriate, but the explanation of robustness checks and endogeneity tests could be expanded for clearer understanding.

Response 3:

We strongly agree with your viewpoint. In the revised manuscript, we have added explanations for robustness checks and endogeneity tests, which can be found in Subsections 4.4 and 4.5, respectively.

Comment 4:

Moderate language editing is required to simplify sentence structure and ensure clarity.

Response 4:

Thanks for your suggestions. We have thoroughly reviewed the entire paper, refining the language and simplifying the sentence structures to enhance clarity and improve readability.

Comment 5:

Suggestions for future research are lacking.

Response 5:

Your feedback is very constructive. In response, we have added Section 5 "Discussion" and Section 6 "Conclusion, Implications, and Limitations," where we suggest directions for future research: first, future studies could broaden their scope to include enterprises from various countries; second, future studies could extend the timeframe of their samples to determine the durability of these conclusions over time.

In summary, we would like to thank you for all your time involved and for this excellent opportunity for us to improve the manuscript. We hope you will find this revised version satisfactory.

Sincerely,

Qian Binhua

Zhejiang business technology institute, Ningbo, Zhejiang Province, China

---

## [Editor Report · Decision Letter 1]

29 Nov 2024

Cost Stickiness, Absorbed Slack and Enterprise Risks: Evidence from China

PONE-D-24-28797R1

Dear Dr. Qian,

We’re pleased to inform you that your manuscript has been judged scientifically suitable for publication and will be formally accepted for publication once it meets all outstanding technical requirements.

Kind regards,

Chen-Wei Yang

Academic Editor

PLOS ONE

Additional Editor Comments (optional):

The revised manuscript demonstrates significant improvements, and the authors’ responses are comprehensive and constructive. The paper is recommended for acceptance for publication.

---

## [Editor Report · Acceptance letter]

4 Dec 2024

PONE-D-24-28797R1 

PLOS ONE

Dear Dr. Binhua, 

I'm pleased to inform you that your manuscript has been deemed suitable for publication in PLOS ONE. Congratulations! Your manuscript is now being handed over to our production team.

Kind regards, 

on behalf of

Professor Chen-Wei Yang 

Academic Editor

PLOS ONE